# Sequential Endoluminal Doxorubicin and Gemcitabine Alternating Weekly with Sequential Mitomycin and Docetaxel for Recurrent Non-Muscle Invasive Urothelial Carcinoma

**DOI:** 10.3390/cancers16244126

**Published:** 2024-12-10

**Authors:** Ian M. McElree, Vignesh T. Packiam, Ryan L. Steinberg, Helen Y. Hougen, Mohamad Abou Chakra, Sarah L. Mott, Michael A. O’Donnell

**Affiliations:** 1Carver College of Medicine, University of Iowa, Iowa City, IA 52242, USA; 2Division of Urology, Department of Surgery, Rutgers Cancer Institute of New Jersey, New Brunswick, NJ 08901, USA; vignesh.packiam@rutgers.edu; 3Department of Urology, University of Iowa, Iowa City, IA 52242, USA; ryan-steinberg@uiowa.edu (R.L.S.); helen-hougen@uiowa.edu (H.Y.H.); mohamad-abouchakra@uiowa.edu (M.A.C.); michael-odonnell@uiowa.edu (M.A.O.); 4Holden Comprehensive Cancer Center, University of Iowa, Iowa City, IA 52242, USA; sarah-mott@uiowa.edu

**Keywords:** urinary bladder neoplasms, chemotherapy, gemcitabine

## Abstract

After failing multiple treatments for non-muscle invasive bladder cancer (NMIBC), patients with this disease face the prospect of radical bladder removal. However, many patients will wish to pursue additional bladder-sparing therapies or lack candidacy for such an invasive procedure. We aimed to report on a tertiary intravesical regimen of sequential doxorubicin and gemcitabine alternating weekly with docetaxel and mitomycin (Quad Chemo) for patients with high-risk NMIBC failing multiple previous treatments. We found in a population of 29 patients with 39 treated units (bladder and/or upper urinary tracts), 80% of the units were disease-free at 3 months and 43% were disease-free at 2 years following Quad Chemo treatment. However, disease progression was a concern with an estimated 43% of patients experiencing disease progression at 5 years. Lastly, side effects were not uncommon with 19 (66%) patients reporting any side effect and 7 (24%) ultimately stopping Quad Chemo due to side effects.

## 1. Introduction

After failure of first-line treatment with Bacillus Calmette–Guérin (BCG), patients with non-muscle invasive urothelial carcinoma (NMIUC) face an increased risk of subsequent disease recurrence and progression [1]. For those meeting FDA-defined BCG-unresponsive status, current American Urological Association (AUA) guidelines recommend offering radical cystectomy, a procedure with significant short- and long-term morbidity while also negatively impacting quality of life [2,3,4,5,6]. However, many patients will either elect to pursue additional bladder-sparing therapy prior to committing to radical cystectomy or lack candidacy for such an invasive procedure.

Despite the emergence of several FDA-approved therapies for BCG-unresponsive NMIUC, the long-term efficacy of these agents has been disappointing [7,8,9,10]. Specifically, the KEYNOTE-057 study investigating pembrolizumab for BCG-unresponsive CIS resulted in a 2-year HGRFS of only 11% [7]. The investigators of nadofaragene firadenovec recently provided long-term results describing a 5-year HGRFS of merely 5.8% in patients presenting with recurrent CIS [9]. The results of nogapendekin alfa inbakicept combined with BCG have been more promising with a 53% 2-year RFS in patients with BCG-unresponsive CIS with or without papillary disease; however, the long-term outcomes remain unclear [10]. Given the often short interval to disease relapse for patients with recurrent CIS, it is crucial for treating urologists to have a range of treatment options available.

As has been demonstrated in the management of metastatic urothelial carcinoma (UC), multi-agent treatment regimens are particularly advantageous for those with multiple treatment failures as they likely better address the complex resistance mechanisms of recurrent UC [11]. At our institution, we have developed and adopted a unique pathway of salvage treatment regimens tailored for those with recurrent UC. We have previously described a high degree of treatment efficacy and low rates of oncological progression when using combinatory treatment regimens for NMIUC, including sequential intravesical gemcitabine and docetaxel (Gem/Doce), valrubicin and docetaxel (Val/Doce), and gemcitabine and cabazitaxel with concurrent intravenous pembrolizumab (GCP) [12,13,14,15]. However, prior to the development of GCP and for those unable to receive GCP due to insurance considerations, our practice has offered a tertiary salvage intravesical treatment modelled after the AG-ITP (doxorubicin, gemcitabine—ifofsamide, paclitaxel, cisplatin) regimen developed at Memorial Sloan Kettering Cancer Center (MSKCC) for patients with metastatic UC [16]. Herein, we describe this regimen of sequential endoluminal doxorubicin and gemcitabine alternating weekly with sequential endoluminal docetaxel and mitomycin (Quad Chemo or AG-DM) for patients with recurrent high-risk NMIUC who were opposed to or unfit for radical surgery.

## 2. Materials and Methods

### 2.1. Study Design and Population

After obtaining IRB approval, we retrospectively reviewed all patients with NMIUC treated with Quad Chemo between December 2007 and April 2024. Patients were included if intending to receive 8 weekly endoluminal instillations of sequential doxorubicin and gemcitabine alternating weekly with sequential docetaxel and mitomycin. Prior to induction, patients with visible tumor received complete resection or ablation. High-risk status was as defined by American Urological Association (AUA) criteria [3,17]. Patients were excluded if they did not undergo at least one follow-up surveillance visit.

### 2.2. Endoluminal Quadruple Chemotherapy Treatment

Patients were sequentially treated with 50 mg of doxorubicin in 50 mL of normal saline for 90 min followed by 1000 mg of gemcitabine dissolved in 50 mL of normal saline for 90 min. This alternated weekly with sequential 37.5 mg of docetaxel in 50 mL of normal saline for 90 min followed by 40 mg mitomycin-C in 20 mL of sterile water for 90 min (Figure 1). Patients repeated this process for a total of 8 weekly treatments. If receiving treatment to the upper tract, each treatment drug was hung 30–35 cm above the flank level and set at a constant gravity rate of 1 drip every 3–4 s for a total of 90 min through a previously placed percutaneous nephrostomy tube as previously described [18]. Patients were instructed to refrain from urinating for 60–120 min following Quad Chemo instillation. Prior to therapy, patients were pre-treated with 2% lidocaine in 50 mL of water with 4 mL of 8.4% bicarbonate for approximately 20 min as local bladder anesthetic and to alkalize the urine, respectively. Maintenance therapy was initiated for up to 2 years until unacceptable toxicities or progression occurred. Maintenance Quad Chemo instillation procedures and dosages matched those used in the induction protocol.

### 2.3. Surveillance

Cancer surveillance took place approximately 4 weeks after ending the 3-month induction period and typically involved formal restaging under anesthesia. Formal restaging procedures included cystoscopy with blue light, bladder barbotage cytology, bilateral upper tract barbotage cytologies, bilateral retrograde pyelograms, random bladder biopsies, and prostatic urethral biopsies. Routine UT investigations (CT, MRI, RGs, URS) were obtained every 6 months for the first 2 years with URS annually, at minimum, for those with documented upper tract disease. If disease-free, repeat surveillance cystoscopy with bladder cytology and FISH was performed quarterly for 2 years, and biannually afterwards.

### 2.4. Analysis

Data including patient clinicopathologic features, treatment history, adverse events (AEs), and oncologic outcomes were retrospectively reviewed and analyzed. The primary outcome was high grade recurrence-free survival (HGRFS). Recurrence was defined as HG tumor relapse in the bladder/upper tract, or prostatic urethra in males. Secondary outcomes included duration of response (DOR), progression-free survival (PFS), cancer-specific survival (CSS), and overall survival (OS). Adverse events (AEs) were determined retrospectively from clinical notes reported by the administering advanced practice provider, who were universally instructed to report any patient-reported side effects at each treatment visit, which were classified by National Cancer Institute Common Terminology Criteria for Adverse Events (CTCAE) version 5. Patients were defined as being intolerant to Quad Chemo if the treatment course was stopped due to symptoms or any serious AE. Survival probabilities were plotted using the Kaplan–Meier method. Estimates along with 95% pointwise confidence intervals were reported. HGRFS was defined as time from initiation of Quad Chemo induction to recurrence. Among patients without recurrence at the initial surveillance, DOR was defined as time from initial surveillance to recurrence. PFS was defined as the time from earliest initiation of Quad Chemo induction to progression defined as T2+ disease, nodal or distant metastasis, or death due to cancer. Otherwise, patients were censored at last disease evaluation. CSS and OS were defined as time from the earliest initiation of Quad Chemo induction to death due to cancer or any cause, respectively. Patients still alive were censored at last follow-up. All statistical testing was two-sided and assessed for significance at the 5% level using SAS v9.4 (SAS Institute, Cary, NC, USA).

## 3. Results

### 3.1. Clinicopathological Characteristics

In total, 29 patients (39 treated units; 26 lower urinary tract, 13 upper urinary tract) with HG NMIUC were included in the final analysis (Table 1). Within the final analysis, 21 (54%) units presented with biopsy-proven CIS of the bladder with or without papillary disease, and another 17 (44%) units presented with presumed CIS in the setting of positive HG cytology in the absence of any tumor. Of the 26 treated lower urinary tracts, 10 (38%) had a history of urothelial carcinoma of the prostatic urethra and/or ducts (PUC), and 4 (15%) presented with PUC immediately prior to Quad Chemo induction. Transurethral resection of prostate (TURP) was performed in 8/10 patients with any history of PUC, including all 4 patients presenting with PUC immediately prior to Quad Chemo induction. The median number of prior inductions was four, with nearly all patients previously failing BCG (96%) and sequential endoluminal doublet chemotherapy regimens (100%). All eligible patients not experiencing treatment-limiting toxicity went on to receive maintenance therapy with a median of 10 (IQR: 5–12) monthly treatments.

### 3.2. Tolerance

In total, 19 (66%) patients reported AEs following treatment (Table 2). There were a total of 37 AEs reported throughout treatment; 16 were grade 1, 18 were grade 2, and 3 were grade 3. The three grade 3 AEs included one case of bladder stone formation, one case of hydronephrosis, and one case of a bladder ulcer secondary to lidocaine toxicity used as a topical bladder premedication. A total of seven (24%) patients stopped Quad Chemo due to side effects, three during induction and four during maintenance. Discontinuation of treatment during induction was secondary to persistent hematuria and frequency symptoms in two out of three patients and lidocaine toxicity in one out of three. During maintenance, two out of four developed chemical cystitis and the other two out of four developed end-stage bladder symptoms. There were no deaths associated with Quad Chemo treatment.

### 3.3. High-Grade Recurrence-Free Survival

Among the treated lower urinary tracts, there were seventeen HG recurrences during follow-up including eleven CIS, three TaHG + CIS, two T1HG, and one positive HG cytology (Figure 2). Of the 14 patients with any component of CIS at time of recurrence, 8 (57%) had disease present within the prostatic urethra. Of the ten patients with a history of PUC, eight recurred during follow-up (six with recurrent PUC).

Concerning the treated upper urinary tracts, there were six recurrences detected via positive high-grade urine cytology. An additional three upper tracts developed suspicious upper tract cytologies during treatment; two out of three received reinduction Quad Chemo and one out of three received systemic pembrolizumab.

Among all of the treated units, complete response (CR) at first surveillance was 79%, and HGRFS was 60% (CI: 43–74%) and 43% (CI: 26–59%) at 1 and 2 years, respectively (Figure 3). Of those disease-free at first surveillance, the median duration of response was 22 months. Subgroup analysis of the lower urinary tracts demonstrated a CR of 80% and HGRFS of 63% (CI: 41–79%) and 38% (CI: 19–58%) at 1 and 2 years, respectively (Appendix A).

### 3.4. Progression and Survival

Median follow-up for survival was 50 (IQR: 39–68) months. During the study period, seven patients received cystectomy (six radical and one partial) and one patient received nephroureterectomy (Appendix A). In total, three out of seven patients had PUC present at the time of cystectomy. The one patient pursuing partial cystectomy developed gross hematuria and an MRI urogram revealed an exophytic mass at the urachal insertion site on the bladder dome with non-specific right external iliac lymph node enlargement. Biopsy of the mass was performed, which revealed high-grade urothelial carcinoma, and, after multidisciplinary tumor board evaluation, the patient was recommended to undergo partial cystectomy with adjuvant nivolumab. The one patient in the upper tract treatment group pursuing nephroureterectomy developed a suspicious cytology and went on to receive additional unit-sparing therapy before developing a definitive HG cytology and electing for surgery.

In total, ten patients experienced disease progression, including four cases of muscle invasive bladder cancer and six cases of metastasis (five distant, one node positive at time of cystectomy). All five patients developing distant metastasis succumbed to a cancer-related death; there were no other cancer-related deaths recorded. The one patient with local node positivity went on to receive adjuvant nivolumab and remained disease-free at the conclusion of the study period. Progression-free survival at 1, 3, and 5 years was 93%, 80%, and 57%, respectively (Figure 4). Cancer-specific survival at 1, 3, and 5 years was 96%, 93%, and 83%, respectively (Appendix A). Overall survival at 1, 3, and 5 years was 97%, 82%, and 69%, respectively (Appendix A).

## 4. Discussion

Our study has several important findings. First, in a heavily pretreated cohort with a median of three prior treatments, Quad Chemo was able to rescue a significant portion of patients with a 79% 3-month complete response rate and an estimated 43% HGRFS at 2 years. By the conclusion of the study period, 10 patients had experienced disease progression and 5 succumbed to a bladder cancer-related death, yielding a 5-year PFS of 57% and CSS of 83%. Lastly, while the majority of AEs were transient and resolved following the end of treatment, 24% of patients ultimately stopped treatment secondary to side effects. These findings indicate that Quad Chemo is a safe and effective salvage therapy for patients with recurrent NMIBC while also highlighting the high-risk nature of these patients, as indicated by the increased risk of disease progression and mortality within the cohort.

The management of urothelial carcinoma has seen significant changes over the past several years. Previously, after failing BCG, no highly effective second-line agent existed and most patients were likely to proceed to radical cystectomy for a non-invasive disease. However, the number of treatment options has expanded considerably with recent advancements in antibody–drug conjugates, immune checkpoint inhibitors, gene therapies, and combinatory chemotherapy regimens [7,8,13,14,19]. In order to provide patients with effective treatment options for NMIUC, our group has utilized the information gained from the study of metastatic UC [16]. In this setting, the chemosensitive properties of UC were immediately apparent after the introduction of systemic MVAC (methotrexate, vinblastine, doxorubicin, and cisplatin) in the 1980s [20]. Subsequently, taxane-containing drug regimens were introduced and assisted in reducing toxic side effects while improving disease-free survival rates [21]. One such combination, AG-ITP (doxorubicin, gemcitabine—ifofsamide, paclitaxel, cisplatin) was developed and later, after dropping the ifofsamide, became the basis for the development of Quad Chemo (doxorubicin, gemcitabine—docetaxel, mitomycin) [16]. Ultimately, the efficacy and widespread availability of combinatory chemotherapy has translated into a successful treatment approach for improving the oncological outcomes of patients with bladder cancer.

Disease progression is of particular concern when treating patients with recurrent cancer unresponsive to prior intravesical therapy. Thus, AUA guidelines recommend offering radical cystectomy to all patients with BCG-unresponsive NMIBC [3]. However, a recently published study found that patients treated with additional bladder-sparing therapy after reaching BCG-unresponsive status had comparable outcomes to those treated with upfront radical cystectomy [22]. However, rates of progression in the bladder-sparing therapy group tended to increase over time relative to those receiving upfront surgery [22]. These findings indicate that there likely exists a window of opportunity to explore additional bladder-sparing therapy while the oncologic risk of progression remains low. Within the current study, treated units had a median of three prior inductions with an estimated 43% of patients experiencing disease progression by 5 years. Further insight into the risk factors and characteristics of those who progress will become increasingly useful as the number of patients pursuing bladder-sparing therapy grows.

The optimal management of patients with non-invasive PUC is not well-established but is an important consideration when combatting recurrent NMIUC. A review of conservative treatment of PUC using intravesical BCG reported complete response rates in the range of 47–72% [22]. However, the authors note that long-term follow-up of these patients results in high rates of cystectomy and disease progression [22]. A phase I trial exploring the use of intravesical cabazitaxel, gemcitabine, and cisplatin (CGC) in patients with recurrent NMIBC found that 3/18 patients experienced disease recurrence solely within the prostatic urethra [23]. Similarly, our group recently reported on the use of sequential intravesical gemcitabine and cabazitaxel with concurrent intravenous pembrolizumab (GCP) as a treatment strategy for patients with recurrent UC of the bladder with or without prostatic urethral involvement [15]. Of the eight patients in this study with a history of PUC, four experienced disease recurrence and two eventually pursued radical cystectomy. Aside from the use of systemic agents like pembrolizumab to control extravesical relapse in the prostatic urethra, some authors have proposed utilizing transurethral resection of the prostate and bladder neck. These procedures offer the benefit of ruling out prostatic stromal disease while eliminating existing PUC and facilitating bathing of the prostatic tissue during intravesical instillations [24,25,26]. The poor outcomes and relatively high incidence of patients with recurrent NMIUC developing PUC highlight the need for more effective treatment strategies in the population.

Radical cystectomy remains the gold standard treatment for patients with recurrent high-risk NMIUC. Quad Chemo presents a promising tertiary treatment option for those who are contraindicated for radical surgery due to frailty, advanced age, or significant comorbidity and/or who desire bladder preservation. While there are many clinical benefits of avoiding invasive surgery, our study highlights the inherent risk of progression and mortality associated with continued bladder preservation in patients with high-risk NMIUC after multiple prior treatment failures. As the number of patients pursuing unit-sparing therapies continues to grow, further research into biomarkers capable of predicting treatment response will optimize patient selection and counseling.

Our study has several limitations. The retrospective nature of the study is prone to selection bias and limits the generalizability of the results presented here. Factors such as healthcare access, referral patterns, and loss to follow-up may misrepresent these results from the broader population of patients with recurrent NMIBC. Additionally, the retrospective assigning of AEs within the medical record may not be fully representative of the true toxicity profile of Quad Chemo treatment. The small cohort size limited the statistical power to detect statistically meaningful associations, and the single-arm design of this study is unable to detect differences between treatment dosages and combinations. Lastly, the patients within this study were treated at a single institution with rigorous NMIUC surveillance and treatment protocols which may not be fully replicated in lower volume centers. Ultimately, while this retrospective report allowed for an exploratory analysis of Quad Chemo, future prospective study would enable a more robust evaluation of this treatment regimen.

## 5. Conclusions

In this retrospective review of 29 patients (39 treated units) with recurrent UC of the upper and lower urinary tracts, we found that Quad Chemo rescued a significant portion of units from disease relapse. However, progression events were considerable, highlighting the high-risk nature of this population. Further prospective evaluation of this treatment regimen is warranted.

## Figures and Tables

**Figure 1 cancers-16-04126-f001:**
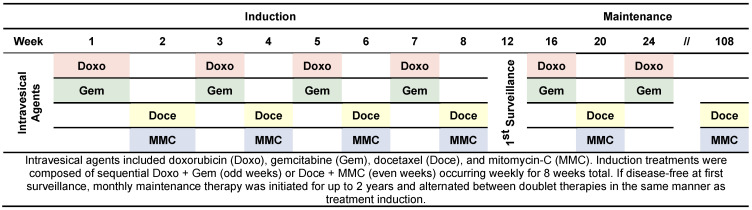
Quad Chemo treatment timeline.

**Figure 2 cancers-16-04126-f002:**
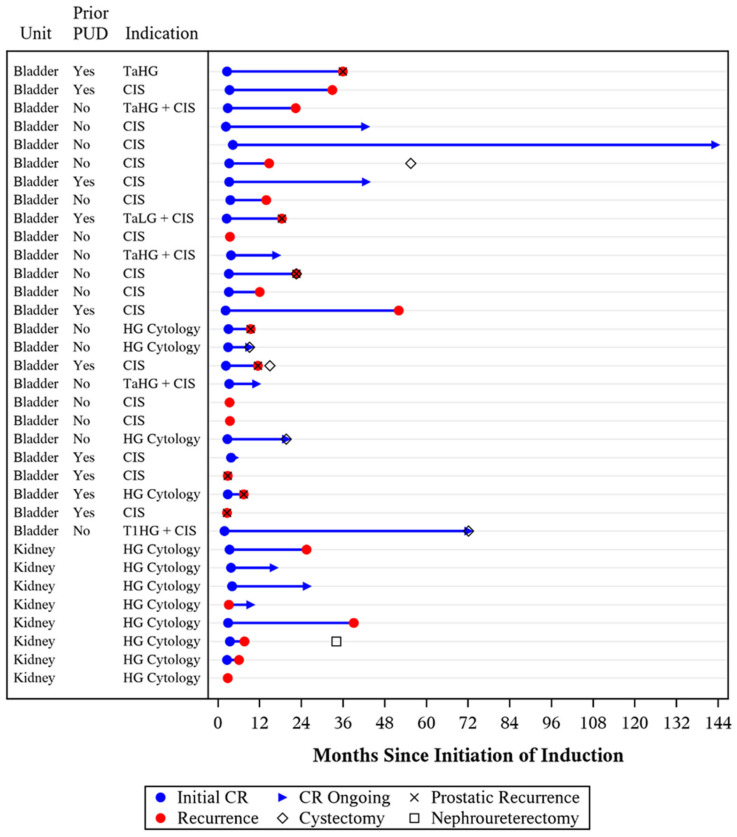
Outcomes of 39 units receiving Quad Chemo treatment.

**Figure 3 cancers-16-04126-f003:**
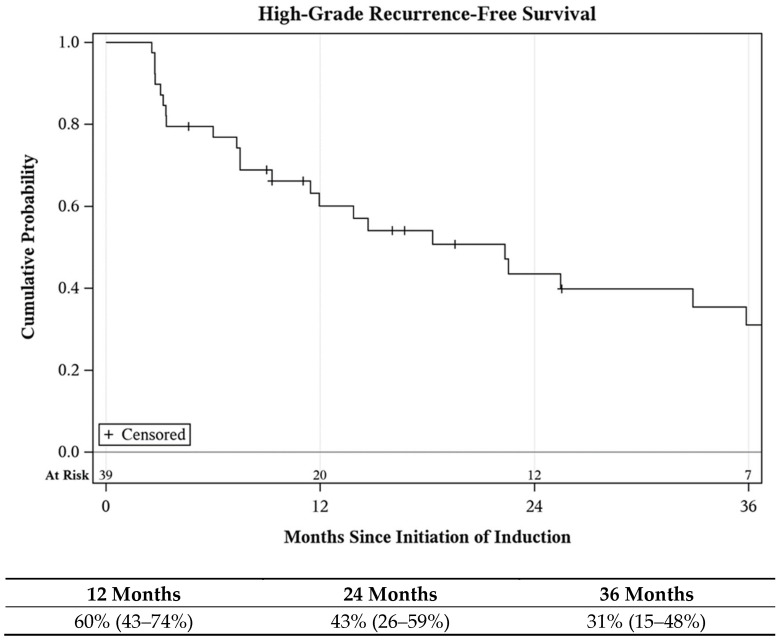
High-grade recurrence-free survival following Quad Chemo treatment.

**Figure 4 cancers-16-04126-f004:**
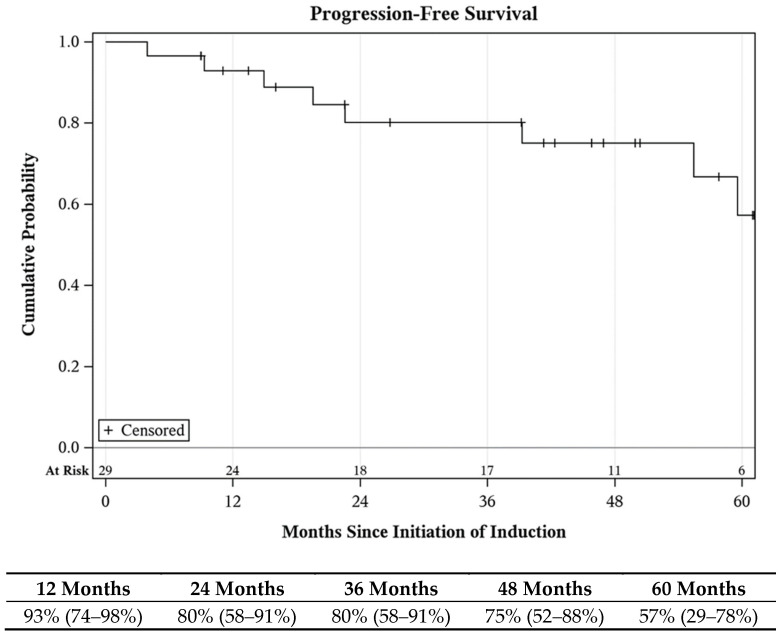
Progression-free survival following Quad Chemo treatment.

**Table 1 cancers-16-04126-t001:** Clinicopathological characteristics prior to Quad Chemo treatments.

		Urinary Tract Location	
	Total	Lower	Upper
No. Patients (%)	29	26	8
Median Age	77 (68–82)	77 (67–81)	82 (78–84)
Sex (%)			
Female	3 (10)	2 (7.7)	1 (13)
Male	26 (90)	24 (92)	7 (88)
Race (%)			
White, Non-Hispanic	28 (100)	25 (100)	8 (100)
Missing	1	1	0
Median ASA Classification (IQR)	3 (2–3)	3 (2–3)	3 (2–3)
Smoking Status			
Current	2 (6.9)	2 (7.7)	0
Former	17 (59)	17 (65)	3 (38)
Never	10 (35)	7 (27)	5 (63)
No. Treated Units (%)	39	26	13
Pre-Treatment Indication (%)			
CIS		16 (62)	-
T1HG + CIS		1 (3.8)	-
TaHG		1 (3.8)	-
TaHG + CIS		3 (12)	-
TaLG + CIS		1 (3.8)	-
High-Grade Cytology		4 (15)	13 (100)
Prior Prostatic Urethral CIS		10 (38)	-
Median No. of Prior Inductions (IQR)		4 (3–4)	3 (2–3)
Median No. of prior BCG courses *		1 (1–2)	0 (0–1)
Prior Treatments (%)			
BCG		25 (96)	6 (46)
Sequential Chemotherapy Regimen **		26 (100)	9 (69)
No. Patients Receiving Maintenance Treatments		18 (769	10 (77)
Median No. of Maintenance Treatments (IQR)		10 (5–12)	5 (3–6)

* Of those previously receiving BCG; ** sequential chemotherapy regimens included gemcitabine and docetaxel, valrubicin and docetaxel, gemcitabine and mitomycin-C, and doxorubicin and docetaxel.

**Table 2 cancers-16-04126-t002:** Adverse events in 29 patients following Quadruple Chemotherapy.

	Grade 1	Grade 2	Grade 3	Total
Total Reported Adverse Events (%)	16	18	3	37
Frequency/Urgency	5 (17)	4 (14)	-	9 (31)
Dysuria	5 (17)	-	-	5 (17)
Bladder Spasm	1 (3)	3 (10)	-	4 (14)
Abdominal/Flank Pain	-	5 (17)	-	5 (17)
Hematuria	2 (7)	2 (7)	-	4 (14)
Nausea/Vomiting	3 (10)	-	-	3 (10)
UTI	-	2 (7)	-	2 (7)
Chemical Cystitis	-	2 (7)	-	2 (7)
Bladder Ulcer	-	-	1 (3)	1 (3)
Hydronephrosis	-	-	1 (3)	1 (3)
Bladder Stone	-	-	1 (3)	1 (3)
Patients Reporting Adverse Events (%)				19 (66)

Adverse events are assigned according to CTCAE V5. Percentages are based upon number of patients receiving treatment.

## Data Availability

Data included in this study are not publicly available.

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
