# Peer review of "Sequential Endoluminal Doxorubicin and Gemcitabine Alternating Weekly with Sequential Mitomycin and Docetaxel for Recurrent Non-Muscle Invasive Urothelial Carcinoma"

_cancers, 2024, doi:10.3390/cancers16244126_

Round 1
Reviewer 1 Report
Comments and Suggestions for Authors
- This study assesses Quad Chemo—a sequential regimen of doxorubicin, gemcitabine, docetaxel, and mitomycin—as a bladder-sparing treatment for recurrent, high-risk NMIUC. It shows a 79% initial response and 43% two-year recurrence-free survival, suggesting efficacy but underscoring a need for further validation.
-
The introduction is clear, but consider briefly discussing alternative bladder-sparing therapies to enhance the reader's understanding of the gap that Quad Chemo aims to fill.
-
While the retrospective design is appropriate for the exploratory nature of this study, discussing potential selection biases more explicitly would enhance the transparency of the study’s limitations. A future prospective study design could also be proposed in the discussion for validation.
-
The description of the chemotherapy protocol is comprehensive. However, adding a flowchart or timeline for treatment stages (induction, maintenance, and surveillance) would improve clarity for readers unfamiliar with the regimen structure.
-
Retrospective AE reporting is inherently prone to underreporting. Clarifying if AE grading followed a standardized reporting framework (such as Clavien Dindo or CTCAE) across all cases would enhance the robustness of toxicity data.
-
The discussion is well written but could benefit from a specific section on implications for clinical practice, especially for patients contraindicated for radical cystectomy. Expanding on factors that might predict better responses to Quad Chemo could also be valuable.
Reviewer 2 Report
Comments and Suggestions for Authors
This is an interesting analysis of patients with BCG unresponsive NMIBC who received multiagent chemotherapy. It should be mentioned that currently novel therapies including systemic immunotherapy and 2 new intravesical agents have been approved in this realm, moreover, radical cystectomy remains the gold standard of care for patients for these therapies are not available.
Abstract:
- Background: It should be mentioned that the effective salvage therapies particularly for the population in hand (heavily treated bladders) the next line of treatment remains radical cystectomy.
- Conclusions: I think the conclusion is not appropriate to the study's findings, for this cohort of population with NMIBC with PFS of 57% and treatment discontinuation of 24% shows that this treatment may not be appropriate for this group of patients.
Manuscript:
Methods:
- Study design – Can the authors elaborate on why the protocol was chosen to be given the way it was given? Meaning why Gemcitabine and docetaxel were not given together?
- Statistical analysis and drug administration are appropriate and clearly described.
Results:
- While this is a retrospective analysis key baseline characteristics of the cohort should be mentioned e.g., ASA or ECOG score, previous intravesical regimens, number of BCG courses, etc.
- For the patients with prior prostatic urethra involvement was TURP performed prior to giving intravesical therapy?
- The presentation of both upper tract and lower tract systems together adds confusion to the results.
- According to Figure 1 it is not clear why patients with continuous CR received radical cystectomy.
Discussion/ conclusions:
- Overall, the PFS, metastases rate, and death appear high for this group of patients that could have been salvaged by radical cystectomy, hence in the discussion I disagree that the current studied regimen is an effective salvage therapy for patients with recurrent NMIBC.
- The main limitations of the study remains the cohort size, single-arm design, and the inherent selection bias coupled with the retrospective nature
